# Design and Experiment of Capacitive Rice Online Moisture Detection Device

**DOI:** 10.3390/s23125753

**Published:** 2023-06-20

**Authors:** Wensheng Sun, Lin Wan, Gang Che, Ping Xu, Hongchao Wang, Tianqi Qu

**Affiliations:** 1Department of Engineering, Heilongjiang Bayi Agricultural University, Daqing 163319, China; 2Heilongjiang Provincial Key Laboratory of Intelligent Agricultural Machinery Equipment, Daqing 163319, China

**Keywords:** rice, moisture content, electric capacity type, genetic algorithms, design, test

## Abstract

To solve the problems of poor stability and low monitoring precision in the online detection of rice moisture in the drying tower, we designed an online detection device for rice moisture at the outlet of the drying tower. The structure of a tri-plate capacitor was adopted, and the electrostatic field of the tri-plate capacitor was simulated using COMSOL software. A central composite design of three factors and five levels was carried out with the thickness, spacing, and area of the plates as the influencing factors and the capacitance-specific sensitivity as the test index. This device was composed of a dynamic acquisition device and a detection system. The dynamic sampling device was found to achieve dynamic continuous sampling and static intermittent measurements of rice using a ten-shaped leaf plate structure. The hardware circuit of the inspection system with STM32F407ZGT6 as the main control chip was designed to realize stable communication between the master and slave computers. Additionally, an optimized BP neural network prediction model based on the genetic algorithm was established using the MATLAB software. Indoor static and dynamic verification tests were also carried out. The results showed that the optimal plate structure parameter combination includes a plate thickness of 1 mm, plate spacing of 100 mm, and relative area of 18,000.069 mm^2^ while satisfying the mechanical design and practical application needs of the device. The structure of the BP neural network was 2-90-1, the length of individual code in the genetic algorithm was 361, and the prediction model was trained 765 times to obtain a minimum *MSE* value of 1.9683 × 10^−5^, which was lower than that of the unoptimized BP neural network with an *MSE* of 7.1215 × 10^−4^. The mean relative error of the device was 1.44% under the static test and 2.103% under the dynamic test, which met the accuracy requirements for the design of the device.

## 1. Introduction

Rice is one of the most important food crops in the world. In China, rice growing areas are mainly distributed in the northern producing areas represented by Heilongjiang Province and the southern producing areas represented by Jiangsu and Anhui Province [1,2,3,4]. Grain drying is an important step to ensure safe food storage. After harvest, rice generally has a moisture content (MC) of 17~25% and needs to be dried using specialized equipment to reach a safe value (standard storage moisture of 14.5%) before storage [5,6]. Bound water and free water are the two physical states of water in rice [7,8]. Generally speaking, free water refers to water that can flow freely in rice and acts as a solvent. Bound water is an important part of the cell structure in rice. In the drying process, if the rice moisture at the outlet of the drying tower cannot be accurately and stably detected, the rice becomes over-dried, and evaporation of bound water in the rice will affect the edible quality of the rice. A lack of drying leaves excess free water in the rice, which will lead to mildew, insects, and other phenomena in the storage of rice [9,10]. These phenomena, in turn, lead to a waste of food and drying energy. Therefore, it is of great importance to design a detection device to improve the online detection accuracy and stability of rice moisture content at the outlet of the drying tower.

At present, grain moisture detection methods are divided into offline detection and online detection methods [11]. Offline detection [12] involves removing the water in grain after the test sample is taken from production and processing and calculating the moisture content of the grain through drying or chemical methods, such as the 105 constant weight method [13] and Karl Fischer method [14]. This type of detection is generally used as the comparison standard for detection accuracy but is not practical in the actual production process. Online detection directly uses equipment in the process of grain production, harvesting, and processing for real-time grain moisture detection. This type of detection is convenient, time-saving, and can meet the needs of actual production. The most widely used online detection methods are the capacitance method and resistance method. The resistance method generally requires the sample to be ground before detection and is a type of lossy detection [15,16,17].

The capacitance method has simple structure, high reliability, good dynamic response, and is economical and simple; this method can easily realize the rapid detection of material moisture [18,19]. Wang et al. [20] designed an online detection system for rice moisture content in a combine harvester by using the capacitance method. The accuracy and stability of rice sample detection were improved by building a sampling platform with the function of secondary screening and impurity removal. Wan et al. [21] took parallel plate capacitors as their research subject, adopted the finned double plate detection method to optimize the rice moisture content detection device, and verified that the detection performance of the device met the actual requirements of rice drying production. Using an impedance analyzer and self-made coaxial cylindrical capacitor, Zhang et al. [22] measured the relative dielectric constant and dielectric loss factor of rice with different water contents at a frequency of 1 kHz~1 MHz. Chen et al. [23] developed a high frequency capacitive online monitoring device for grain moisture content. This device can monitor grain moisture content online and display and store it in real time. Zhang et al. [24] designed a concentric circle plane capacitance moisture measuring instrument for corn grain combines based on dielectric properties. This instrument was found to satisfy the accuracy of grain moisture measurement and shorten the measurement time. Li et al. [25] designed an online grain moisture detection device based on the principle of capacitance detection. For wheat samples with moisture content of 10~25%, the average relative error between the measured value and actual moisture content was less than 1%. The rice test value of the DM510 grain dryer produced by DANTEC in Canada was less than 0.5% [26].

The capacitance method is easily affected by environmental factors and edge effects when detecting rice moisture. Thus, the detection accuracy and stability of this method cannot meet the actual working requirements of the drying tower. In addition, the bipolar plate parallel capacitor structure is used most commonly, while the tripolar plate capacitor structure is rarely used for research on rice moisture detection. For this reason, the present study adopted the triple plate parallel plate detection mode and designed plate structure parameters through simulation and experimentation. The online rice moisture content detection device was composed of a dynamic rice collection device and detection system. A prediction model for a BP neural network optimized using the genetic algorithm was established to correct the master computer moisture value. Through the structural design of a capacitive sensor, the design of an online detection device, and the establishment of a prediction model, a capacitive online rice moisture detection device was designed, which provides a reference for the innovative study and development of rice moisture online detection technology and devices.

## 2. Materials

The device was composed of a dynamic rice collection device and real-time detection system. The detection system included hardware design and software design. In order to realize the online detection mode of dynamic continuous sampling and static intermittent measurements, it was necessary to design a dynamic collection device to realize the running process of periodic sampling, detection, and abandonment of rice samples flowing out of the grain outlet, as well as a real-time detection system to realize rice moisture and temperature signal acquisition, processing, the output of detection results, and storage.

### 2.1. Design of Dynamic Rice Collection Device

The dynamic collection device was composed of a servo, power switch, coupling, smooth shaft, flat key, bearing seat, ten-shaped leaf plate, acrylic plate, angle iron, vibration device, circuit board package shell, grain storage bench, and other parts, as shown in Figure 1. In order to smoothly connect the device and drying tower test bench grain discharge mouth, the length, width, height, and thickness of the sampling box were designed to be 209, 182, 500, and 3.6 mm, respectively, with an acrylic plate used for splicing. The angle iron was fastened with bolts, and the size of the rice sampling cavity was 200 × 180 × 110 mm. In order to achieve the necessary size of the device and realize the meshing requirements of the electrode plate, a ten-shaped leaf plate structure was designed, which was achieved via 3D printing and made of high-toughness resin.

Through the ten-shaped leaf plate structure, the dynamic collection device’s sampling and grain discharge functions were realized. During the drying operation of the drying tower, the rice flowed from the grain discharge mouth of the drying tower into the sampling cavity of the dynamic collection device after cleaning. During the sampling stage, the three electrode plates were kept parallel, and the servo did not function. After the rice filled the sampling cavity, capacitive and temperature sensors collected the rice sample signals. After the signals were collected, the timer’s first period was completed, the servo drove the coupling to make the ten-shaped leaf plate start to flip 180°, the device drained the grain, and the rice flowed from the sampling cavity down the slope into the grain storage bench via gravity. At the end of grain discharge, the timer’s second period was completed, the servo stopped working, the ten-shaped leaf plate completed its 180° rotation, and the other surface of the leaf plate received rice from the grain discharge mouth for the next signal acquisition. The above actions were then completed to ensure that the device periodically sampled, measured, and discarded the samples and completed rice sample measurement to update the set of monitoring data. Table 1 shows the main component information.

### 2.2. Hardware Design

The Altium Designer software was used to design the rice moisture content detection circuit, with STM32F407ZGT6 as the core processing chip. The operation amplification circuit, temperature sensor, power supply regulation and conversion circuit, and communication module were selected as the main modules of the circuit. The design of the hardware system is shown in Figure 2.

#### 2.2.1. Detection Unit

Capacitance detection;

In this system, capacitor detection used a parallel capacitor structure with three plates. The plate material was copper foil, which offers excellent conductivity, good surface smoothness, good ductility, and a low price. According to the results in the optimization experiment for the structural parameters of the tri-plate capacitor, the length, width, and thickness of the side plates were set as 180, 100, and 1 mm, respectively, the spacing of the plates was chosen to be 100 mm, and the outside of the copper foil was brushed with insulating paint, which had both anti-wear and electrostatic shielding properties.

Since the change in rice moisture content between plates had little effect on the capacitance, it corresponded to the detection of small capacitance signals [27,28]. In this study, the operational amplification circuit was selected as the detection circuit of the capacitive sensor and consisted of a Miller integral circuit composed of 1B and a Schmitt circuit composed of 2B. The two amplifier models were LM741 and LM709, and the circuit voltage was ±12 V. The operational amplifier circuit diagram is shown in Figure 3a.

Temperature detection;

The DS18B20 digital temperature sensor was used in this system for temperature sensing. The sensor communicated through a single bus serial port, with a temperature operating range of −55 to +125 °C. In the range of −10 to +85 °C, the detection accuracy could reach ±0.5 °C, which met the testing requirements. The schematic diagram of temperature measurement circuit is shown in Figure 3b.

#### 2.2.2. Power Supply Unit

The system was powered by switching the power supply of 24 V DC through the WRB2412MD chip to process 24 into 12 V DC to supply power to the capacitive sensor. This power then ran through the LM7805 buck chip to convert 12 into 5 V DC in order to power the microcontroller module and then again through the LM1117-3.3 integrated voltage regulator chip through DC–DC conversion to power the 3.3 V communication unit. The circuit diagram of power supply unit is shown in Figure 3c.

#### 2.2.3. Communication Unit

The communication function between the slave and the master computers was realized via RS485 serial communication. The power supply voltage of the communication unit was 3 V, and the optocoupler was added to isolate the input and output signals from each other. There were three independent interrupt sources for receiving completion interrupts, sending completion interrupts, and registering null interrupts. The 6LB184 was used as a communication unit transceiver with integrated transient voltage rejection and a data transfer rate of 250 kbps. The circuit diagram of communication unit is shown in Figure 3d.

### 2.3. Software Design

The master computer was based on the Visual Basic development platform for program development programming. The interface had a serial port module, configuration module, device address module, moisture calibration module, temperature calibration module to set the master computer, and real-time display of the rice moisture and temperature information; the right line chart could draw moisture, temperature, and capacitance value curves. The interface diagram of the master computer is shown in Figure 4.

Using the Keil uVision 5 development environment, the C language was used to write the software for the slave computer to realize the initialization settings of various modules, registers, serial ports, and sensor data acquisition and preprocessing in the system. The system flowchart is shown in Figure 5.

## 3. Methods

### 3.1. Detection Principle

At room temperature, the relative medium of water is about 80, and the relative medium of dry rice is about 2.5. When rice passed through the capacitor, the difference of moisture content led to a change in the relative dielectric constant between the capacitor plates, leading to a change in capacitance values [29,30,31].

Ignoring the edge effect, the capacitance *C* of the flat plate capacitor is expressed as follows [32]:(1)$$C=\frac{\varepsilon A^{\prime }}{l}=\frac{\varepsilon_{r}\varepsilon_{0}A^{\prime }}{l}$$
where *A*’ is the covered area of the capacitor’s bipolar plates; *Ɛ* is the dielectric constant of the medium; *Ɛ*_r_ is the relative permittivity of the medium; *Ɛ*_0_ is the permittivity of the vacuum; and *l* is the distance between plates.

The medium between the three-plate flat capacitor (assuming that the three plates have the same size) is simplified to rice dry matter (solid), water (liquid), and air (gas). The equivalent model is shown in Figure 6.

Assume that the stacking shape of the three mediums is cuboid with the same height and length *l*, and the spacing *d* between the upper plates is the same as that between the lower plates. Due to symmetry, the derivation of the relationship between the MC and capacitance between the upper plates is the same as that below. To simplify the derivation process, we derived the correlation formula only between the upper plates.

According to the capacitor parallel connection formula, since the voltage values between the plates were the same, the capacitance value *C*_up_ between the upper plates is the sum of the capacitance values of the three media:(2)$$C_{up}=\frac{\varepsilon_{0}W}{d}\left(\frac{A_{1}}{W}\varepsilon_{1}+\frac{A_{2}}{W}\varepsilon_{2}+\frac{A_{3}}{W}\varepsilon_{3}\right)$$
where *Ɛ*_1_ is the accumulation of relative permittivity of the rice dry matter (solid state); *Ɛ*_2_ is the relative dielectric constant of water (liquid); *Ɛ*_3_ is the relative dielectric constant of air (gas); *A*_1_, *A*_2_, and *A*_3_ are the effective area of rice dry matter (solid), water (liquid), and air (gas) accumulation between the upper plates, respectively; *d* is the distance between the upper plates; *W* is the sum of the effective widths of the three accumulated media between the upper plates.

In the capacitive sensor, *A* and *d* are constants as follows:(3)$$K_{0}=\frac{\varepsilon_{0}A}{d}$$

We define the void ratio of rice between the upper two plates *e*_1_ as
(4)$$e_{1}=A_{3}/A$$

According to the definition of rice moisture content, ignoring the air quality, the MC *M*_up_ between the upper two plates can be deduced as
(5)$$M_{up}=\frac{\rho_{2}V_{2}}{\rho_{1}V_{1}+\rho_{2}V_{2}}\times 100\%=\frac{\rho_{2}W_{2}}{\rho_{1}W_{1}+\rho_{2}W_{2}}$$
where *ρ*_1_ is dry matter density of rice; *ρ*_2_ is water density; *V*_1_ is the volume of rice dry matter accumulation between the upper plates; *V*_2_ is the volume of water accumulation body between the upper plates; *W*_1_ is the effective width of rice dry matter accumulation between the upper plates; and *W*_2_ is the effective width of the water accumulation body between the upper plates.

The *M*_up_ can be calculated by Equations (2)–(5):(6)$$M_{up}=\frac{1}{\frac{\left[C_{up}-\varepsilon_{2}\cdot K_{0}-\left(\varepsilon_{3}-\varepsilon_{2}\right)e_{1}\cdot K_{0}\right]\cdot \rho_{1}}{\left[K_{0}\left(\varepsilon_{1}-\varepsilon_{2}\right)\left(1-e_{1}\right)\right]-\left[C_{up}-\varepsilon_{2}\cdot K_{0}-\left(\varepsilon_{3}-\varepsilon_{2}\right)e_{1}\cdot K_{0}\right]}+1}$$
(7)$$M=\frac{\left(M_{up}+M_{down}\right)}{2}$$
where *M* is the moisture content of rice in the capacitor, and *M*_down_ is the moisture content of rice between the lower plates.

### 3.2. Electrostatic Field Simulation of the Tri-Plate Capacitor

If the parallel plates are approximately infinite, the electric field formed between the plates is a well-proportioned electric field, and the internal electric field is uniformly distributed [33,34]. However, the size of the parallel plates was actually limited, so the electric field was not an ideal uniform electric field. At the fringe of the capacitor plate, the field lines of the electric field were curved and emanative, resulting in edge capacitance of the parallel plate, making the actual capacitance of the capacitor greater than the theoretical capacitance [35,36,37]. It was found that the thickness [38,39,40], spacing, and relative area of parallel plate capacitors were the main factors affecting the edge effect. In this section, the COMSOL Multiphysics 5.6 software was used to simulate the three-dimensional electric field of the parallel-plate condenser, and the control variable method was used to conduct a single factor analysis of the plate thickness, plate spacing, and relative area to explore their relationship. The model modeling and meshing effects are shown in Figure 7.

#### 3.2.1. Electrostatic Field Analysis of Capacitor Plate Thickness

Here, the electrode plate spacing *d* remained 100 mm, the electrode plate length was 150 mm, and the relative area *A* was 15,000 mm^2^. Simulation research was carried out for thickness *h* values of 0.5, 1.0, 1.5, 2.0, 2.5, and 3 mm. The distribution of electric field lines generated by different plate thicknesses is shown in Figure 8.

In Figure 8, the white arrow represents the direction of electric field flow, and the red line represents the electric field line. With an increase in plate thickness, the electric field line at the plate fringe increasingly expanded and diverged, and the distribution density of the electric field line became larger. This result shows that with a plate thickness of 0.5~3 mm, the influence degree of the edge effect strengthens with an increase in plate thickness.

#### 3.2.2. Electrostatic Field Analysis of Capacitance Plate Spacing

The electrode plate thickness *h* was maintained at 0.5 mm, the electrode plate length was 150 mm, and the relative area *A* was 15,000 mm^2^. Simulation research was carried out with distance *d* between the plates set to 80, 85, 90, 95, 100, and 105 mm. The distribution of electric field lines generated by different plate spacing is shown in Figure 9.

As shown in Figure 9, in the range of 80–105 mm between the plate spacing, the scattering distribution density of electric field lines at the edge of the capacitor increased with an increase in plate spacing, indicating that the influence degree of the edge effect increased with an increase in plate spacing.

#### 3.2.3. Electrostatic Field Analysis of the Relative Area of Capacitor Plates

Maintaining the plate thickness *h* at 0.5 mm, the plate spacing *d* at 100 mm, and the plate width at 100 mm, we adjusted the relative area size by changing the plate length and taking the relative area *A* as 15,000, 18,000, 21,000, 24,000, 27,000, and 30,000 mm^2^ for the simulation. The different relative plate areas for the distribution of the electric field line are shown in Figure 10.

As shown in Figure 10, when the length of the plate increased, the electric field line at the fringe of the plate shrank to the middle plate, and the density of the electric field line became increasingly lower, indicating that the influence degree of the edge effect in the range of 15,000~30,000 mm^2^ of the plate area weakened with an increase in the relative area.

Combined with the simulation analysis results and actual device conditions, the parameter range of the plate can be determined as follows: plate thickness of 0.5~2.5 mm, plate spacing of 80~100 mm, and relative area of 18,000~30,000 mm^2^.

### 3.3. Optimizing the BP Neural Network Prediction Model with the Genetic Algorithm

#### 3.3.1. Data Preparation

Japonica rice in the Daqing region was prepared as the test sample after screening and filtering to restore the real conditions for rice moisture measurements in the drying tower. Considering the temperature conditions during the rice harvest season and the actual range of rice moisture content at the outlet after drying in the drying tower, the moisture range of the rice samples was determined to be 10~26%, and the temperature range was 16~25 °C. Rice samples with moisture content from 10 to 26% were prepared via the drying method at 105 °C, with each group corresponding to 1%, for a total of 16 groups, and 2 kg of rice samples prepared in each group. The debugged detection device was then placed in a constant temperature and humidity box from 16 to 25 °C, with 1 °C increments as a temperature gradient. The prepared rice sample was poured into the sampling chamber, and the capacitance difference between the plates before and after placement of the rice sample was measured using a capacitance measuring instrument. Each group test was repeated 5 times, producing 800 sets of data in total.

#### 3.3.2. Model Building

Based on genetic algorithm (GA) and BP neural network theory [41,42,43,44,45,46], the programming design of the BP neural network was optimized via the GA using the MATLAB language in the MATLAB software. The structure of the BP neural network was 2-90-1, with 270 weights and 91 thresholds. The individual coding length of the genetic algorithm was set to 361.

In total, 800 sets of data prepared from the rice samples were used as input and output data for the GA–BP neural network, and 560 sets were randomly selected as training data, 120 as validation data, and 120 as test data. The schematic diagram of the BP neural network structure is shown in Figure 11.

#### 3.3.3. Evaluation Index

In order to measure the prediction effect of the prediction model on the moisture content of rice, the regression model goodness-of-fit index R and mean squared error (*MSE*) were used as evaluation indicators.

Goodness-of-fit index R calculation formula:(8)$$R=1-\sqrt{\frac{\sum_{i=1}^{n}{\left(z_{i}-P_{i}\right)}^{2}}{\sum_{i=1}^{n}z_{i}^{2}}}$$
where *P*_i_ is the fitted value; *z*_i_ is the response value; and *n* is the sample size.

*MSE* calculation formula:(9)$$MSE=\frac{1}{n}\sum_{i=1}^{n}{\left(x_{i}-y_{i}\right)}^{2}$$
where *x*_i_ is the predicted value, and *y*_i_ is the actual value.

The closer the index R is to 1, the smaller the *MSE* value becomes, indicating that the prediction model has a better effect on the prediction of rice moisture content.

## 4. Results and Discussion

### 4.1. Optimization Experiment on the Structural Parameters of the Tri-Plate Capacitor

Japonica rice in the Daqing area was selected as the test sample, and the MC of the rice samples was prepared using the 105 °C constant weight method [47,48]. We placed quantitative samples into an aluminum box after drying to a constant weight and weighed them. The weight was recorded as *m*_1_. Then, we put the samples into a 105 °C blast drying oven at a constant temperature to re-bake them to constant weight; the result was recorded as *m*_2_. The formula for calculating rice moisture content *M* is as follows:(10)$$M=\frac{m_{1}-m_{2}}{m_{1}}\times 100\%$$
where *m*_1_ is the rice quality before drying, and *m*_2_ is the rice quality after drying.

The test factor code table is shown in Table 2.

The test took the plate thickness *h*/mm, plate spacing *d*/mm, and relative area *A*/mm^2^ as the factors and the ratio of the measured capacitance value *C*_1_ of the capacitor to the theoretical calculated capacitance value *C*_2_ (sensitivity *Y*) of the capacitance meter as the test index to carry out the three-factor five-level central composite design. The test scheme and results are shown in Table 3.

The experimental data were processed using the Design-Expert 12 software to obtain a variety of fitting model comparisons, as shown in Table 4.

The ANOVA for the quadratic model is shown in Table 5.

It can be seen from Table 4 that the cubic equation is complex and unnecessary. By comparing the size of the *R*^2^ correction values and *R*^2^ prediction values of the linear model, two-factor model, and quadratic model, we found that the larger the *R*^2^ value, the better the model fit. Here, the quadratic equation provided the most reasonable results.

As shown in Table 5, the *p*-value of the model was found to be less than 0.01. Additionally, the lack of fit was not significant, indicating that the regression model was extremely significant, and the equation fitting effect was good. Plate thickness *X*_1_; plate spacing *X*_2_; relative area *X*_3_; and interactions *X*_1_ *X*_2_, *X*_1_ *X*_3_, *X*_2_ *X*_3_, and *X*_3_^2^ showed significant influences, but the other factors were not significant. Excluding the insignificant factors, the fitted equation is as follows:(11)$$Y=10.89520+0.425956X_{1}-0.087810X_{2}-0.000521X_{3}+0.009546X_{1}X_{2}-5.1\times 10^{-5}X_{1}X_{3}+\;3.24056\times 10^{-6}X_{2}X_{3}+6.24468\times 10^{-9}X_{3}^{2}.$$

The response surfaces of the interactions of plate thickness *X*_1_, plate spacing *X*_2,_ and relative area *X*_3_ with sensitivity *Y* are shown in Figure 12. Here, the response surface shapes of the interactions *X*_2_ *X*_3_ and *X*_1_ *X*_3_ to *Y* are concave, and the color difference is large, indicating that *X*_2_ *X*_3_ and *X*_1_ *X*_3_ had a significant effect on *Y*. The response surface of interaction *X*_1_ *X*_2_ to *Y* is relatively flat compared to those of *X*_2_ *X*_3_ and *X*_1_ *X*_3_, but the overall influence on *Y* is significant.

The larger the plate spacing, the larger the sampling cavity of the device, and the more rice samples were measured each time. Under these conditions, the detection time was also relatively reduced. Moreover, the smaller the relative area, the lower the manufacturing cost, the smaller the area of the rice friction plate, and the longer the plate could be used. The middle plate needed to have a certain bearing capacity on the mechanical structure of the detection device, and the thickness of the plate needed to be greater than or equal to 1 mm. The plate size with sensitivity *Y* closest to 1 was the optimal combination of parameters. Based on the above boundary conditions, the model was established as
(12)$$\left\{\begin{array}{c} Y\to 1\\ MinX_{3}\\ MaxX_{2}\\ 1\leq X_{1}\leq 2.5 \end{array}.\right.$$

After software processing, the optimal plate parameter combination was determined to be plate spacing 100 mm, plate thickness 1 mm, and relative area 18,000.069 mm^2^, with a capacitance ratio of 1.056 under this parameter combination. This combination was the least affected by the edge effect.

### 4.2. Analysis of the Prediction Results

The parameters of the BP neural network and genetic algorithm are shown in Table 6 and Table 7.

The training results are shown in Figure 13.

As shown in Figure 13, the *MSE* of the training, validation, and testing continued to decrease with an increase in training iterations. After a total of 771 training iterations, the lowest *MSE* of 1.9683 × 10^−5^ was obtained after 765 iterations, which was lower than that of the unoptimized BP neural network *MSE* value of 7.1215 × 10^−4^, indicating that the prediction accuracy of the BP neural network optimized by the genetic algorithm was higher. To make the test set samples representative, samples were taken every 1% from an MC range of 10~26%, as shown in Figure 13c, which indicates that the predicted values of GA–BP were basically attached to the curve of the actual value. The regression R value of the training set was 0.99998, the regression R value of the validation set was 0.99997, the regression R value of the test set was 0.99997, and the regression R value of all sets was 0.99998, indicating that the model fitting effect was good and can provide a reference for moisture correction in the master computer.

### 4.3. Indoor Accuracy of the Device

The experiment was carried out in the intelligent agricultural machinery equipment laboratory of Heilongjiang Bayi Agricultural University, and japonica rice in the Daqing region was selected as the test sample. The rice samples required for the test were screened and filtered. The conditions of the indoor static test are shown in Figure 14. In order to verify the accuracy error of the device, 19 sets of rice samples were configured using the 105 °C drying method with moisture content of 9.46, 10.16, 11.27, 12.35, 13.46, 14.72, 15.22, 16.79, 17.23, 18.75, 19.56, 20.24, 21.82, 22.43, 23.75, 24.56, 25.12, 26.33, and 27.46%. The designed device and a resistive moisture meter (PT-8188 type) were used for testing, and the measurements were repeated three times for each group to take the average.

As shown in Figure 15, the maximum relative error of the absolute value of the resistive moisture meter was −0.64%, the smallest relative error of the absolute value was 0.40%, and the mean relative error (MRE) was 0.52%. The largest relative error of the absolute value of the device was 1.77%, the smallest relative error of the absolute value was −1.12%, and the MRE was 1.44%. The MRE difference between this device and the resistive moisture meter was less than 1%, which indicates that the device offered high detection accuracy under the static test conditions in the laboratory.

### 4.4. Indoor Dynamic Experiences of the Device

To verify the performance of the device in the indoor bench test, the test was carried out in the intelligent equipment laboratory of Heilongjiang Bayi Agricultural University. The dynamic test conditions are shown in Figure 16. The rice variety used in the experiment was Suijing 18, with the following natural attributes: 12 leaves on the main stem, long-grain type, plant height of about 104 cm, and panicle length of about 18.1 cm. The weight of 1000 grains was about 26.0 g. We installed this device to the grain discharge port of the drying tower bench.

The device in the bench test was operated normally to ensure the periodic updating of rice samples and that the master computer could stably display the moisture of the rice samples in the sampling chamber. As shown in Table 8, the MRE of the detection results of the device was 2.103%, less than 2.5%, which was able to meet the demands for the online detection accuracy of rice moisture in the indoor dynamic test.

### 4.5. Discussion

In this study, an online moisture detection device for rice based on the capacitance method was designed based on four aspects: the design of capacitor structure, the design of a dynamic acquisition device, the design of a real-time detection system, and the establishment of a rice moisture prediction model. The results provide a design reference for the innovative development of a moisture detection device for grain drying operations and moisture detection research in smart agriculture.

In this study, a triode plate with a plate capacitor structure was designed, and the detection principle of the capacitive sensor for rice moisture was analyzed by establishing a mathematical model. In order to reduce the influence of the edge effect on sensor detection accuracy, the COMSOL Multiphysics 5.6 software was used to simulate the electrostatic field of the tri-plate capacitor; single-factor analysis was carried out on the plate thickness, plate spacing, and plate relative area; and a reasonable parameter range was selected to carry out the central composite design of three factors and five levels. The structural parameters of the plate with the lowest influence of the edge effect were determined through the following optimization conditions: plate thickness 1 mm, plate spacing 100 mm, and relative area of 18,000.069 mm^2^.

A reasonable rice moisture content detection model is the key to an accurate detecting device. At present, the conventional method is to use the data obtained from the calibration test in data processing software for regression operations in order to establish the detection model. In this study, model training of the data was carried out using the BP neural network, and the genetic algorithm was used to find the optimal weight and threshold of the BP neural network. Based on the training results, the BP neural network prediction results optimized by the genetic algorithm were more accurate, and the model offered a high goodness-of-fit. However, because the parameters of the genetic algorithm and the BP neural network were constantly adjusted after training, this method has certain subjectivity. Consequently, the prediction accuracy of this model still has a large optimization space. Future research could focus on improving the genetic algorithm; e.g., through the particle swarm algorithm, gray neural network, or other algorithms, to obtain a more high-quality and accurate prediction model.

The device designed in this study was applied to the grain discharge port of the drying tower test bench. By designing a ten-shaped leaf plate structure, the sampling, grain discharge mechanism, and capacitor were combined to improve the operating performance of the device. Through the indoor static test and dynamic test, we verified that the online detection accuracy of the device was high, and the equipment ran well. However, during the testing process, we found that the flow state of rice in the sampling chamber affected the detection accuracy. The influencing factors will be studied in detail in the next stage of this research to further improve the detection accuracy of the device.

## 5. Conclusions

Using the COMSOL Multiphysics 5.6 software, an electrostatic field simulation of a three-plate plate capacitor was carried out, and the parameter range of the plate was determined as follows: plate thickness of 0.5~2.5 mm, plate spacing of 80–100 mm, and relative area of 18,000~30,000 mm^2^. The structural parameters of the capacitive plates were tested, and the optimal structural parameters were found to be 1 mm thickness, 100 mm spacing, and a relative area of 18,000.069 mm^2^. The parameters of the plate were least affected by the edge effect and met the requirements of mechanical design. A dynamic acquisition device with a ten-shaped leaf plate as the key component and a detection circuit with the STM32F407ZGT6 chip as the core were designed. We designed the master computer using the Visual Basic development platform. Using the Keil uVision 5 development environment, development and programming of the slave computer were carried out with the C language, thereby realizing the stable acquisition, processing, visualization, and storage of signals. In this way, we established a prediction model for the genetic algorithm to optimize the BP neural network. The structure of the BP neural network was 2-90-1, the individual coding length of the genetic algorithm was 361, and the lowest value of *MSE* obtained after 765 iterations of prediction model training was 1.9683 × 10^−5^, which was lower than the *MSE* value of 7.1215 × 10^−4^ belonging to the unoptimized BP neural network. Additionally, the regression R values of the training set, validation set, test set, and all sets were good, thus providing a reference for moisture correction in the master computer. The measurement results of the designed capacitive online rice moisture detection device were verified, and the results showed that the MRE of the device was 1.44% under indoor static test conditions and 2.103% under indoor dynamic test conditions. The accuracy and stability of rice moisture detection at the outlet of the drying tower were improved by designing a three-plate capacitive rice moisture detection device online, which provided a rapid measurement method for realizing accurate rice drying.

## Figures and Tables

**Figure 1 sensors-23-05753-f001:**
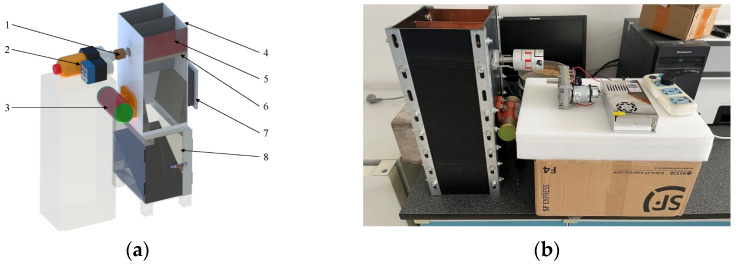
Dynamic collection device: (**a**) 3D modeling diagram; (**b**) physical map: 1. coupling; 2. servo; 3. vibrating device; 4. acrylic plate; 5. copper foil; 6. ten-shaped leaf plate; 7. circuit board package shell; 8. grain storage bench.

**Figure 2 sensors-23-05753-f002:**
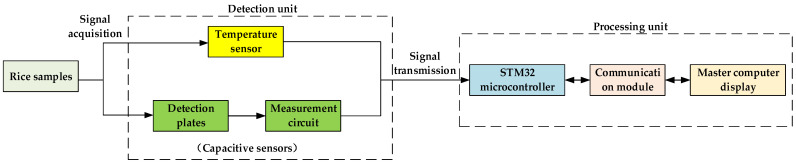
Hardware schematic blueprint.

**Figure 3 sensors-23-05753-f003:**
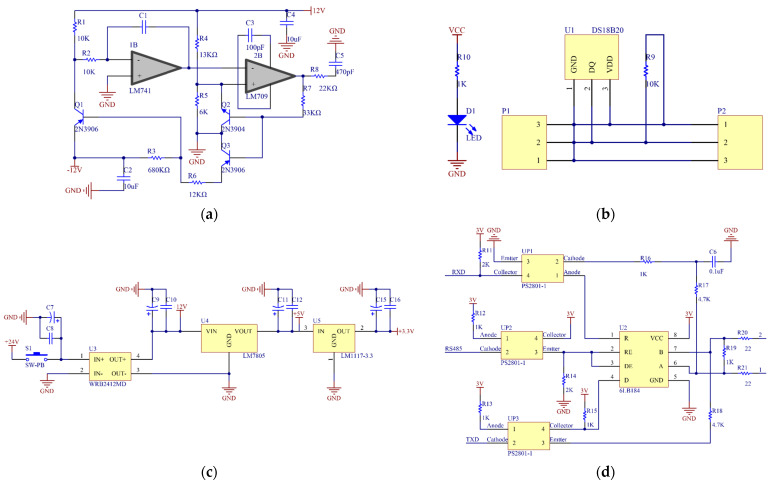
Hardware circuit design: (**a**) operational amplifier circuit diagram; (**b**) schematic diagram of temperature measurement circuit; (**c**) circuit diagram of power supply unit; (**d**) circuit diagram of communication unit.

**Figure 4 sensors-23-05753-f004:**
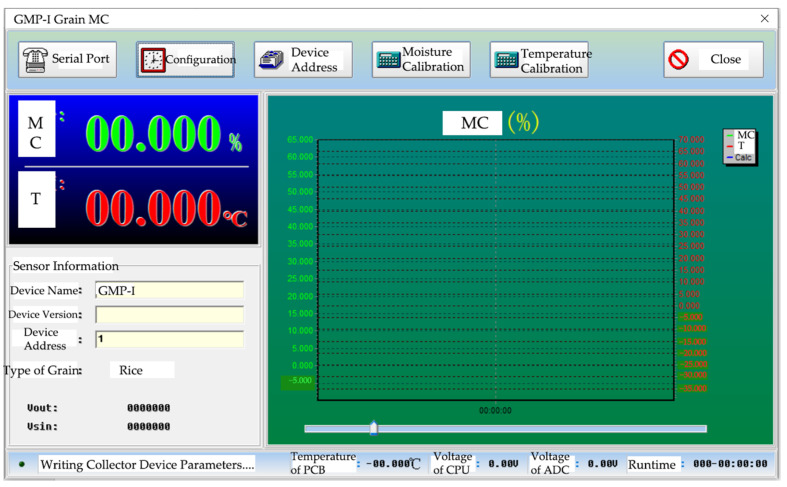
Interface diagram of the master computer.

**Figure 5 sensors-23-05753-f005:**
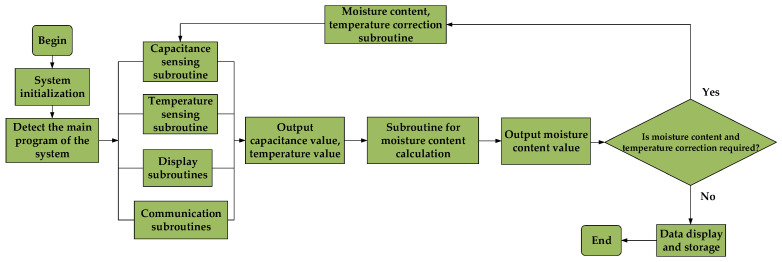
System flowchart.

**Figure 6 sensors-23-05753-f006:**
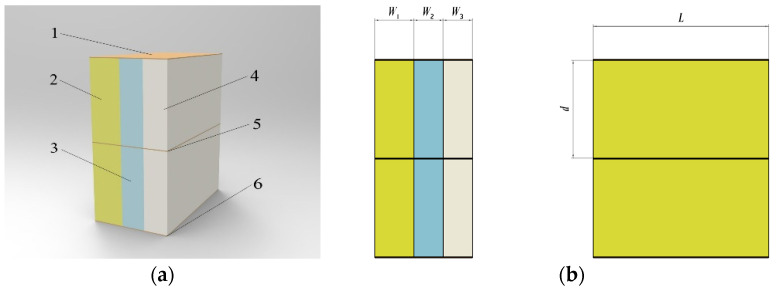
Model of equivalence: (**a**) diagrammatic drawing of the equivalent model; (**b**) schematic diagram of model dimensions. 1. Upper plate; 2. dry rice; 3. water; 4. air; 5. middle plate; 6. lower plate.

**Figure 7 sensors-23-05753-f007:**
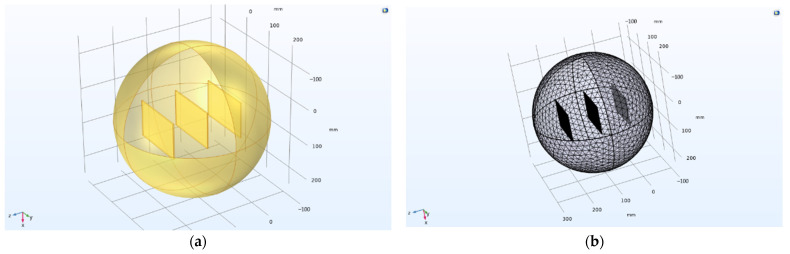
Modeling and gridding: (**a**) model effects; (**b**) meshing effects.

**Figure 8 sensors-23-05753-f008:**
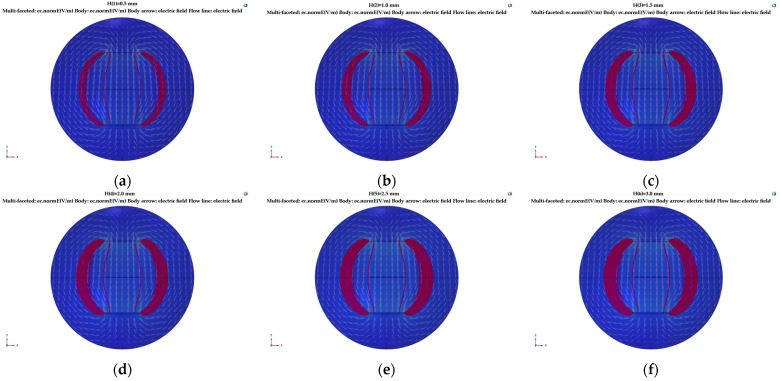
Simulation of the edge scattering electric field with a change in the thickness of the tri-pole plate: (**a**) *h* = 0.5 mm; (**b**) *h* = 1.0 mm; (**c**) *h* = 1.5 mm; (**d**) *h* = 2.0 mm; (**e**) *h* = 2.5 mm; (**f**) *h* = 3.0 mm.

**Figure 9 sensors-23-05753-f009:**
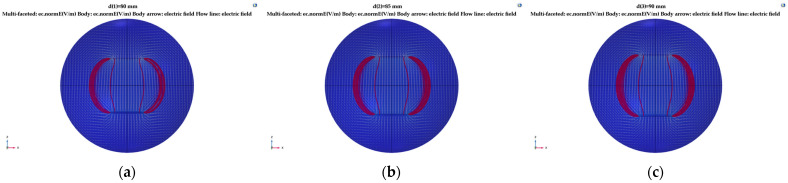
Simulation of the scattering electric field from the edge with variation of the tri-pole spacing: (**a**) *d* =80 mm; (**b**) *d* =85 mm; (**c**) *d* = 90 mm; (**d**) *d* = 95 mm; (**e**) *d* = 100 mm; (**f**) *d* = 105 mm.

**Figure 10 sensors-23-05753-f010:**
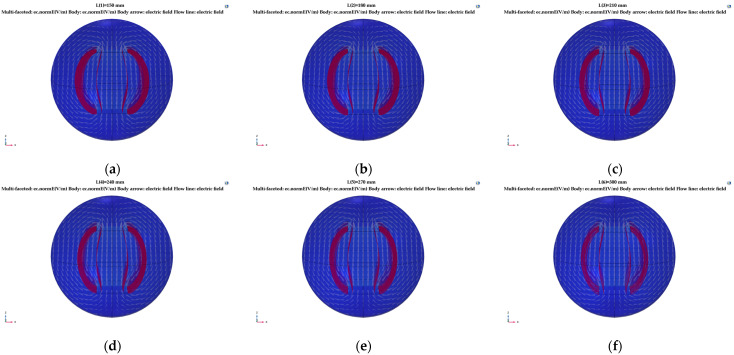
Simulation of the edge scattering electric field with a change in the tri-pole’s relative area: (**a**) *A* = 15,000 mm^2^; (**b**) *A* = 18,000 mm^2^; (**c**) *A* = 21,000 mm^2^; (**d**) *A* = 24,000 mm^2^; (**e**) *A* = 27,000 mm^2^; (**f**) *A* = 30,000 mm^2^.

**Figure 11 sensors-23-05753-f011:**
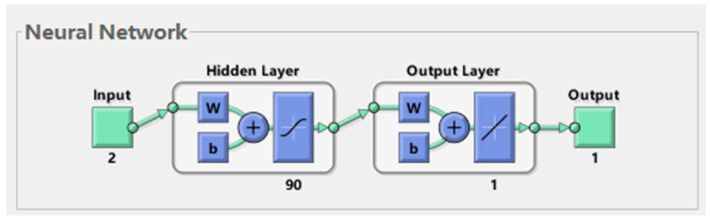
Schematic diagram of the BP neural network structure.

**Figure 12 sensors-23-05753-f012:**
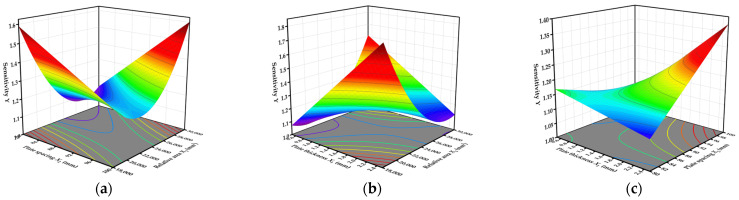
Response surface of each interaction: (**a**) interaction *X*_2_ *X*_3_ response surface; (**b**) interaction *X*_1_ *X*_3_ response surface; (**c**) interaction *X*_1_ *X*_2_ response surface.

**Figure 13 sensors-23-05753-f013:**
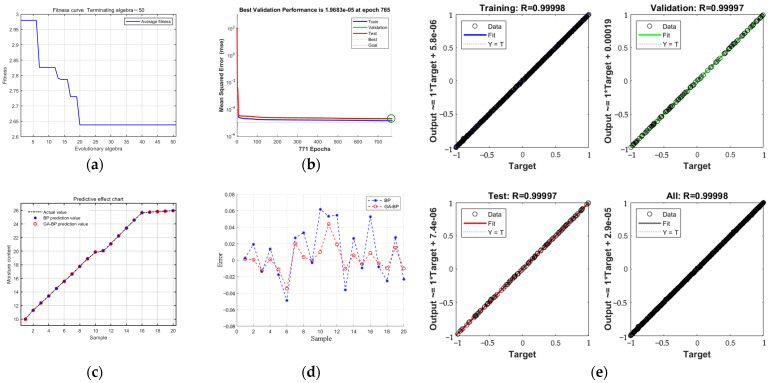
The result from running the algorithm: (**a**) evolutionary process; (**b**) GA–BP prediction model training results; (**c**) GA–BP forecast renderings; (**d**) test sample error comparison; (**e**) regression effect of the GA–BP prediction model.

**Figure 14 sensors-23-05753-f014:**
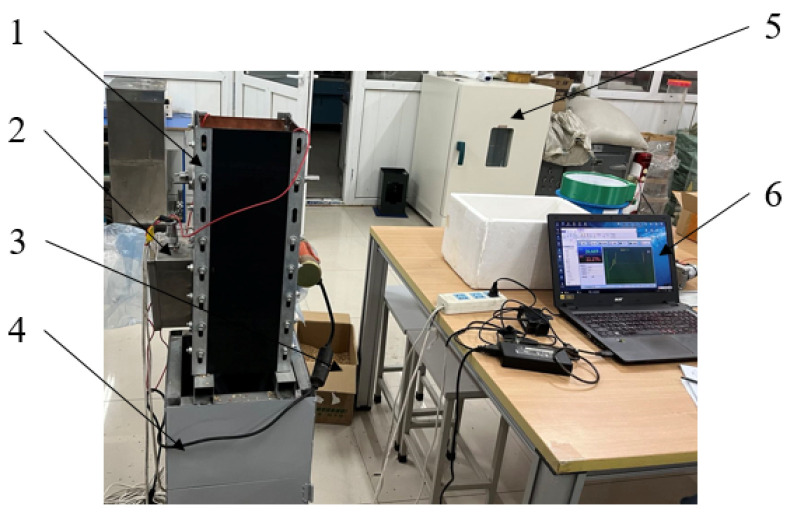
Conditions of the indoor static test: 1. dynamic acquisition device; 2. circuit board package shell; 3. rice before processing; 4. grain storage bench; 5. drying box; 6. master computer.

**Figure 15 sensors-23-05753-f015:**
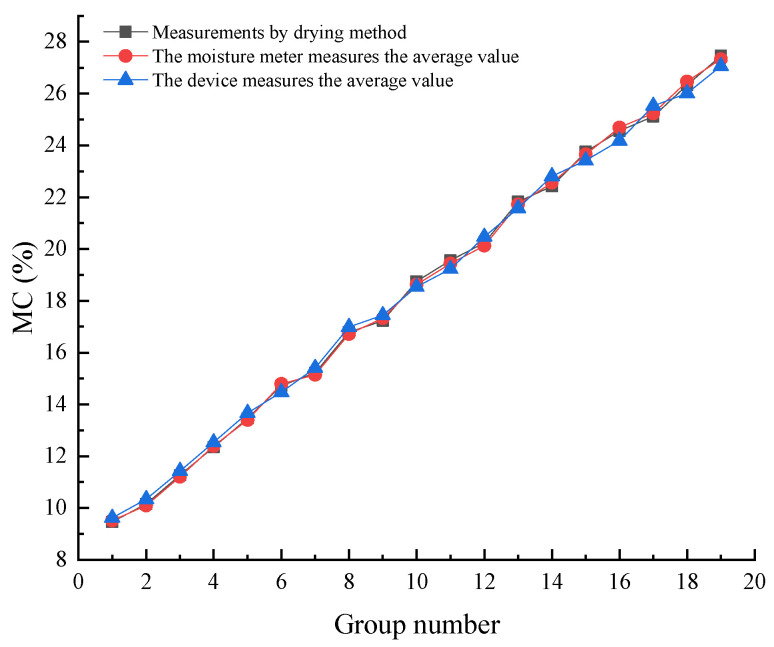
Test results of the moisture content using different methods.

**Figure 16 sensors-23-05753-f016:**
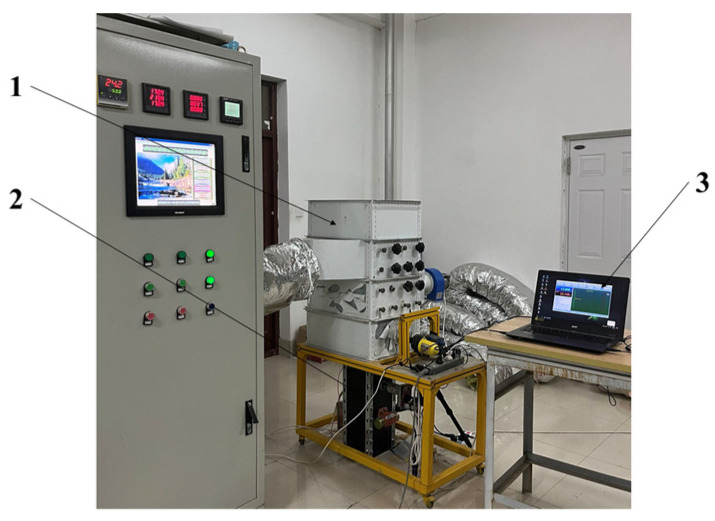
Dynamic test conditions: 1. drying tower; 2. dynamic acquisition device; 3. master computer.

**Table 1 sensors-23-05753-t001:** Parts of the components of the main parameter table.

Name	Main Parameters
Servo	Rated voltage 12~24 V/DC
Power switch	Output voltage: 24 V, AC output current: 10 A, power 250 W
Coupling	Plum type aluminum alloy material Size: Outer diameter 55 mm, length 78 mm, hole 12 to 15 mm
Normal flat key	Type A 6 × 6 × 180 mm, 304 stainless steel
Bearing seat	Rhombic shape KFL004 inner diameter 20 mm
Vibrating device	Single 220 V 20 W
Grain storage bench	Length, width, and height: 300 × 300 × 500 mm
Circuit board package shell	Length, width, and height: 145 × 145 × 75 mm

**Table 2 sensors-23-05753-t002:** Test factor code table.

Coded Value	Plate Thickness *h*/mm	Plate Spacing *d*/mm	Relative Area *A*/mm^2^
−1.682	0.5	80	18,000
−1	0.9	84.1	20,432
0	1.5	90	24,000
1	2.1	96.0	27,568
1.682	2.5	100	30,000

**Table 3 sensors-23-05753-t003:** Test scheme and results.

Test Number	*X* _1_	*X* _2_	*X* _3_	Measured Capacitance Value *C*_1_/pF	Theoretical Calculated Capacitance Value *C*_2_/pF	Sensitivity *Y*
1	0	0	1.682	44.8	33.68	1.33
2	0	0	0	39.63	34.16	1.16
3	0	0	0	38.94	34.16	1.14
4	0	0	0	39.28	34.16	1.15
5	−1	1	−1	37.21	35.1	1.06
6	0	0	−1.682	36.48	25.69	1.42
7	0	−1.682	0	44.31	38.53	1.15
8	1.682	0	0	41.13	33.44	1.23
9	0	0	0	39.39	34.16	1.23
10	0	1.682	0	43.25	35.45	1.22
11	−1.682	0	0	35.91	32.94	1.09
12	1	−1	−1	44.31	31.43	1.41
13	−1	1	1	42.44	29.89	1.42
14	−1	−1	1	44.67	35.74	1.25
15	0	0	0	39.97	34.16	1.17
16	1	1	1	43.22	33.77	1.28
17	0	0	0	38.6	34.16	1.13
18	−1	−1	−1	40.23	32.18	1.25
19	0	0	0	39.63	34.16	1.16
20	1	−1	1	43.48	41.02	1.06
21	0	0	0	41.68	34.16	1.22
22	0	0	0	38.6	34.16	1.13
23	1	1	−1	43.69	30.34	1.44

**Table 4 sensors-23-05753-t004:** Comparison of multiple fitting models.

Source	Std. Dev.	*R* ^2^	Adjusted *R*^2^	Predicted *R*^2^	PRESS
Linear	0.1175	0.1027	−0.0389	−0.5404	0.4503
2FI	0.0869	0.5869	0.4320	−0.1190	0.3271
Quadratic	0.0369	0.9394	0.8974	0.7505	0.0729
Cubic	0.0379	0.9559	0.8921	−0.7651	0.5160

**Table 5 sensors-23-05753-t005:** ANOVA for the quadratic model.

Source	Sum of Squares	df	Mean Square	F-Value	*p*-Value
Model	0.2746	9	0.0305	22.38	<0.0001
*X* _1_	0.0145	1	0.0145	10.66	0.0062
*X* _2_	0.0089	1	0.0089	6.50	0.0243
*X* _3_	0.0066	1	0.0066	4.88	0.0458
*X*_1_ *X*_2_	0.0091	1	0.0091	6.69	0.0226
*X*_1_ *X*_3_	0.0946	1	0.0946	69.41	<0.0001
*X*_2_ *X*_3_	0.0378	1	0.0378	27.74	0.0002
*X* _1_ ^2^	0.0002	1	0.0002	0.1490	0.7057
*X* _2_ ^2^	0.0024	1	0.0024	1.80	0.2031
*X* _3_ ^2^	0.1007	1	0.1007	73.85	<0.0001
Residual	0.0177	13	0.0014		
Lack of Fit	0.0071	5	0.0014	1.07	0.4431
Pure Error	0.0106	8	0.0013		
Cor Total	0.2923	22			

Note: *p* < 0.01 indicates a very significant difference; *p* < 0.05 indicates a significant difference.

**Table 6 sensors-23-05753-t006:** BP neural network parameter settings.

Number of Trainings	Learning Rate	Training Accuracy	Training Function
1000	0.1	0.00001	Levenberg–Marquardt

**Table 7 sensors-23-05753-t007:** Genetic algorithm parameter settings.

Population Size	Evolutionary Algebra	Crossover Probability	Mutation Probability
100	50	0.3	0.1

**Table 8 sensors-23-05753-t008:** Bench test results.

Group Number	Measurements under the Drying Method (%)	Device Measurements (%)	Relative Error (%)
1	18.67	19.03	1.93
2	17.34	17.01	−1.90
3	16.67	17.02	2.10
4	16.21	15.85	−2.22
5	15.93	15.59	−2.13
6	15.63	16.01	2.43
7	15.32	15.02	−1.96
8	15.11	15.41	1.99
9	14.97	14.63	−2.27
10	14.79	14.48	−2.10

## Data Availability

Not applicable.

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
