# Peer review of "Design and Experiment of Capacitive Rice Online Moisture Detection Device"

_sensors, 2023, doi:10.3390/s23125753_

Round 1

Reviewer 1 Report

My suggestions to the authors:

1)      Improve the English grammar usage.

2)      Some of the text and numbers in the figures are too small to easily read.

3)      Please verify that the images presented in Figure 8 are indeed the simulation results from varying the distance d between the plates.

Improve the English grammar usage.

Author Response

Dear Review,

Thank you for your valuable suggestions, which have played an important role in improving the quality of our paper.

For your comments and Suggestions:

  • Improve the English grammar usage.
  • Some of the text and numbers in the figures are too small to easily read.
  • Please verify that the images presented in Figure 8 are indeed the simulation results from varying the distance d between the plates.

We have completed the correction, which are as follows:

  1. This paper uses MDPI language editing service to improve the quality of English content.
  2. The content of Figure 4, Figure 8(original figure 7), Figure 9(original figure 8), Figure 10(original figure 9) has been adjusted to make it clearer.
  3. After checking, the image in Figure 9 (original Figure 8) is the simulation result of changing the plate spacing d, and the simulation analysis results of Figure 9 are restated, "As can be seen from Figure 9, in the range of 75mm~100mm between the plate spacing, the scattering distribution density of electric field lines at the edge of the capacitor increases with the increase of the plate spacing, indicating that the influence degree of the edge effect increases with the increase of the plate spacing".

Thank you again for your review and guidance to this paper, please let us know if you have any revisions, and sincerely hope that our paper can be published in the journal.

Reviewer 2 Report

·        The article's writing needs to improve in its wording and the correct English grammar use.

·        The introduction to the issue being addressed is minimal. It needs to delve deeper into the background of the problem since the information about state of the art about investigations carried out in this field is nonexistent, and the justification for the work needs to be stronger.

·        There needs to be a general justification for the materials used in the research and why they are the most suitable for this investigation.

·        Figures 7 and 8 contain texts that cannot be read. The information should be presented as legibly as possible.

·        In the Discussion section, information that should have been exposed and developed in the Introductory section, and sources that support it are indicated.

·        The presentation of the research results does not represent the expected conclusions from a research paper. They should show how they improve previous models, what the obtained results mean, and the contribution to the research field to which they subscribe.

   The article's writing needs to improve in its wording and the correct English grammar use.

Author Response

Dear Review,

Thank you for your valuable suggestions, which have played an important role in improving the quality of our paper.

For your comments and Suggestions:

  • The article's writing needs to improve in its wording and the correct English grammar use.
  • The introduction to the issue being addressed is minimal. It needs to delve deeper into the background of the problem since the information about state of the art about investigations carried out in this field is nonexistent, and the justification for the work needs to be stronger.
  • There needs to be a general justification for the materials used in the research and why they are the most suitable for this investigation.
  • Figures 7 and 8 contain texts that cannot be read. The information should be presented as legibly as possible.
  • In the Discussion section, information that should have been exposed and developed in the Introductory section, and sources that support it are indicated.
  • The presentation of the research results does not represent the expected conclusions from a research paper. They should show how they improve previous models, what the obtained results mean, and the contribution to the research field to which they subscribe.

We have completed the correction, which are as follows:

  1. This paper uses MDPI language editing service to improve the quality of English content.
  2. The Introduction has been revised. The research background of the paper and the issue being addressed are reorganized. The grain moisture detection methods are divided into offline detection and online detection. Off-line detection generally used as the comparison standard of detection accuracy, but not practical in the actual production process. The widely used online detection methods are capacitance method and resistance method. The resistance method generally requires the sample to be ground before detection, which belongs to lossy detection. The research status of grain online moisture detection technology based on capacitance method was analyzed. At present, the accuracy and stability of rice online moisture detection at the outlet of drying tower are poor, which results in the waste of food and drying energy. Based on this question, an online detection device for rice moisture at the outlet of drying tower was designed.
  3. This justification for the materials used in the research was added to the materials section and explained why they are the most suitable for this investigation. The device is composed of dynamic rice collection device and real-time detection system. The detection system includes hardware design and software design. In order to realize the on-line detection mode of dynamic continuous sampling and static intermittent measurement, it is necessary to design a dynamic collection device to realize the running process of periodic sampling, detection and abandonment of rice samples flowing out of the grain outlet, and design a real-time detection system to realize the functions of rice moisture and temperature signal acquisition, processing, detection results display and storage.
  4. The content of Figure 4, Figure 8(original figure 7), Figure 9(original figure 8), Figure 10(original figure 9) has been adjusted to make it clearer.
  5. The introductory information in the discussion section has been adjusted to the introduction section, the information has been revised, and the relevant supporting sources has been cited.
  6. The content of the Results, Discussions and Conclusions sections has been revised.

Thank you again for your review and guidance to this paper, please let us know if you have any revisions, and sincerely hope that our paper can be published in the journal.

Reviewer 3 Report

 It is aimed at the problems of poor stability and low monitoring precision of online detection of rice moisture in the drying tower, an online detection device for rice moisture at the outlet of drying tower was designed. It could be improved as following:

1.in "2.3. Software Design" part, it should add a flow diagram about the system.

2.in"3.3.1. Data PreparatioRice  "samples with moisture content from 10% to 26% were prepared by drying method at 105°C, each 1% group, a total of 16 groups, and 2kg of rice samples were prepared in each  group. The debugged detection device is placed in a constant temperature and humidity  box, from 16°C to 25°C every 1°C as a temperature gradient, and each group is repeated  5 times, a total of 800 sets of data. It should introduce why choose 10%-26% and 16°C to 25°C. What about other range? and it should give more detail information about the sample range.

3.It should add more recent year references,especially in recent 3 years.

Moderate editing of English language.

Author Response

Dear Review,

Thank you for your valuable suggestions, which have played an important role in improving the quality of our paper.

For your comments and Suggestions:

  • in "2.3. Software Design" part, it should add a flow diagram about the system.
  • in"3.3.1. Data Preparation" rice samples with moisture content from 10% to 26% were prepared by drying method at 105°C, each 1% group, a total of 16 groups, and 2kg of rice samples were prepared in each group. The debugged detection device is placed in a constant temperature and humidity box, from 16°C to 25°C every 1°C as a temperature gradient, and each group is repeated 5 times, a total of 800 sets of data. It should introduce why choose 10%~26% and 16°C to 25°C. What about other range? and it should give more detail information about the sample range.
  • It should add more recent year references, especially in recent 3 years.

We have completed the correction, which are as follows:

  1. In "2.3. Software Design" part, Figure 5 depicts the flow diagram of the system being added.
  2. Considering the temperature conditions during the rice harvest season and the actual range of rice moisture content at the outlet after drying in the drying tower, the moisture range of rice samples was determined to be 10%~26%, and the temperature range was 16°C~25°C. More detailed sample information has been added to the Supplementary file.
  3. The references of the paper have been revised, and a total of 48 references have been made in this paper, including 40 references in the last 3 years.

Thank you again for your review and guidance to this paper, please let us know if you have any revisions, and sincerely hope that our paper can be published in the journal.